# Spatio-Temporal Changes of Land-Use/Land Cover Change and the Effects on Ecosystem Service Values in Derong County, China, from 1992–2018

**Yanru Wang, Xiaojuan Zhang and Peihao Peng \***

College of Earth Sciences, Chengdu University of Technology, Chengdu 610059, China; YanruWang610059@163.com (Y.W.); zxj6120@126.com (X.Z.)

\* Correspondence: pengpeihao@cdut.edu.cn; Tel.: +86-1598-2328-087

**Abstract:** Monitoring the spatio-temporal variation of the land-use/land cover change (LULC) and ecosystem service value (ESV) changes will help achieve regional sustainable development and management. Derong County is a part of the Hengduan Mountains area, the most crucial ecological functional area in China, and LULC has changed tremendously in the past 30 years. However, the effects of LULC changes on ecosystem services is not well understood. Based on 1992, 1995, 2005, 2013, and 2018 remote sensing images, we used visual interpretation to obtain LULC data and used global value coefficients and modified local value coefficients to assess the spatial-temporal changes of ESV and LULC from 1992 to 2018. The results showed that: (1) From 1992 to 2018, shrubland and grassland decreased, while built-up land, snow, forestland, water body, and cropland area increased. (2) The ESV with an overall decrease of $0.25 \times 10^8$ yuan, ecological projects have played a positive role in improving ESV. In contrast, the main decrease factor of ESV was the increase in agricultural economic development and urban expansion from 1992 to 2018. (3) The ESV spatial distribution indicated the value density of ESV was on the decline, and with the greatest deterioration in Dianyagong. The highest density of ESV area is distributed in Waka, and the lowest density of ESV area is distributed in Bari. This research points out the important role of Derong County in the regional life support system and provides a scientific reference for the sustainable management of dry-hot valley regions' land resources and ecosystem services.

**Keywords:** Derong County; ecosystem service value; temporal and spatial variation; land-use and land cover change

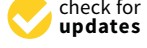

## 1. Introduction

Ecosystem services are closely related to human well-being, are essential to human life, and involve all the benefits that humans obtain from natural ecosystems [1]. In 1997, a global research boom in ecosystem services was set off. Costanza evaluated the ecosystem services value (ESV) in a quantitative manner for the first time and proposed to divide ecosystem services into 17 types, thus laying a foundation for research [1–3]. Ecosystem service values (ESVs) is a monetized form of ecosystem service functions [4,5]. In recent years, the ESV has become a hot research question in ecology. However, the ESV has decreased in many regions, which proposed crucial challenges to human well-being and livelihoods [3,6–10]. Experts and scholars have reached a consensus on the importance of incorporating "ecosystem services" into resource management decisions [11–13]. Policymakers are also paying more and more attention to ESV. Therefore, ESVs research has received enough attention [14,15].

Globally, land-use/land cover change (LULC) is an extremely dominant factor affecting the ESV changes [16–18]. In 1995, since the LULC research plan was jointly proposed by IHDP (International human dimensions program) and IGBP (International geosphere-biosphere program), LULC research has become the frontier and hot issue of global environ-

mental change research [19–21]. LULC, as the most basic human practice, can most immediately mirror the effects of human activities on global change [5,22]. The intensification of LULC leads to changes in the ESVs, which has a profound impact on global environmental change and sustainable development [23–27]. Over the past 60 years, LULC has severely affected the value, benefits, and regional ecological security of ecosystem services [28–31]. Land-use activities have caused dysfunctional ecosystems and unbalanced ecosystem structures, leading to many environmental problems, such as environmental pollution, land degradation, biodiversity decline, and soil erosion [32–34]. Therefore, it is essential to explore the connection between LULC and ESV to reveal the effect of human activities on the ecosystem and coordinate the relationship between humans and nature [21,35–37]. At present, remote sensing (RS) has the advantages of macroscopic, fast, and real-time and is widely used in various fields. The spatial information technology represented by RS and geographic information system (GIS) provides a convenient and effective method for the study of LULC change and the temporal and spatial dynamics of ESV [38].

As a part of the Hengduan Mountains, Derong County is the most representative dry-hot valley region in China [39,40], which is located in the second largest natural forest distribution area and the upper reaches of the Yangtze River. It is an extremely important ecological function area, which plays an important role in regulating climate, balancing carbon dioxide in the atmosphere, conserving water, preventing soil erosion, and protecting the ecological environment in downstream areas [41–43]. Due to agricultural development, overgrazing, and urbanization, a series of protection policies and ecological projects were implemented, and brought about large changes in land-use in Derong county in the past 30 years, especially in dry-hot valley areas [44]. However, the effects of LULC changes in Derong County on ecosystem services are not yet well understood, while it extremely lacks in dry-hot valley regions. Thus, there is an urgent need to carry out this research.

Based on the research gap identified above and remote sensing data, this study used the revised equivalent factor method, quantitatively evaluated the LULC changes, ESV changes and analyzed the effects of LULC on ESV of Derong County from 1992 to 2018. There are two specific goals:

(1) reveal the temporal and spatial variations of the LULC in Derong County from 1992 to 2018;
(2) evaluate the temporal and spatial changes of the ESV in response to LULC.

This research is essential for the rational use, protection, and management of land resources in Derong County, promotion of the sustainable development of ecosystem services, and the realization of the coordinated development of economic and ecological protection. These results will provide a scientific reference for scientifically formulating ecological protection measures in the Hengduan Mountains region and dry-hot valley regions of other countries.

## 2. Materials and Methods

### 2.1. Study Area

Derong County (99°07′ E to 99°34′ E, 28°09′ N to 29°10′ N) located in the southeastern edge of the Qinghai-Tibet Plateau and southwestern part of Sichuan Province, China (Figure 1). It is a principal ecological functional area in China and belongs to the Hengduan Mountains region, the most representative dry-hot valley region in China [39–44]. It belongs to the subtropical arid valley climate zone, with sufficient sunshine [39,40]. The average annual rainfall is 327.1 mm, and the annual average temperature is 14.8 °C. The annual average frost-free period is 243 d, the annual average sunshine time is 2200.7 h, and the annual average relative humidity is 46%. The river in Derong County belongs to the Jinsha River water system.

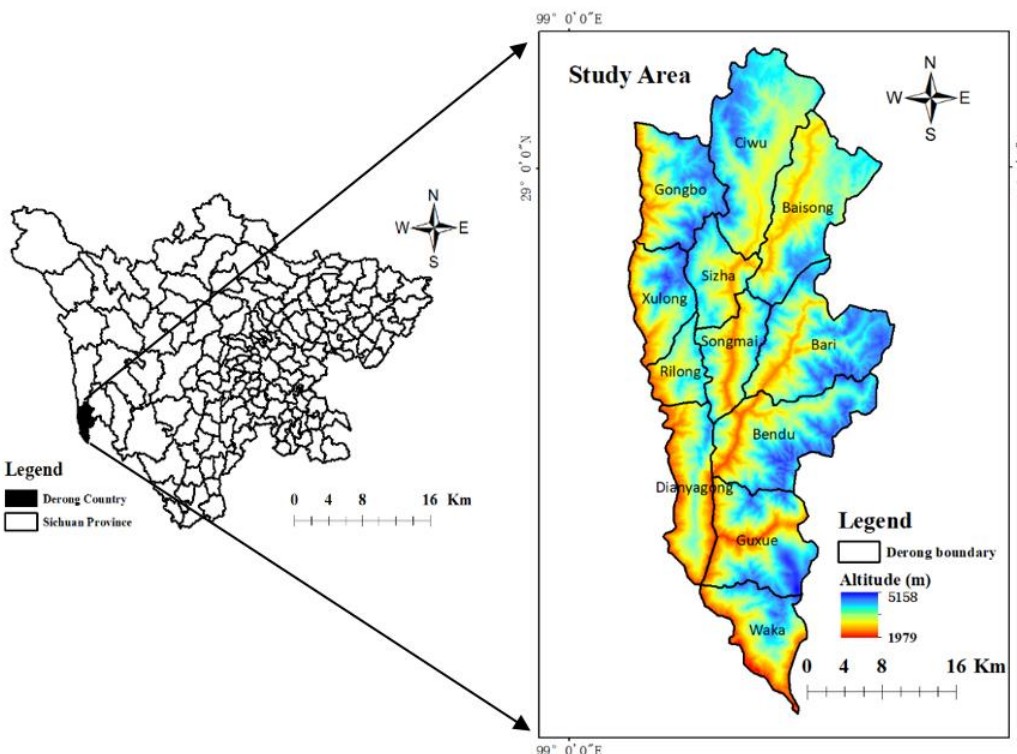

**Figure 1.** Location of the Derong County.

## 2.2. Remote Sensing Data and Data Processing

The Landsat TM/OLI remote sensing image data were derived from the U.S. Geological Survey (https://www.usgs.gov/). The five remote sensing data years were: 1992, 1995, 2005, 2013, and 2018. First, ENVI5.3 software was used to perform the atmospheric correction, geometric correction, band synthesis, image enhancement, and other data preprocessing. According to GB/T 21010-2017 "Land-Use Status Classification," the national ecosystem classification system and the national ecological remote sensing survey classification plan were combined with the natural and economic development characteristics of the Hengduan Mountain region in Derong County [45], the LULC type was divided into seven categories: Forestland, shrubland, grassland, snow, water body, cropland, and built-up land (Figure 2). Then, various maps and data of Derong County were combined, including the ground truth data of Derong County obtained from the Global Positioning System (GPS) field survey, Landsat images, existing maps, and Google Earth images were used as reference data, based on the texture, morphology, and tone characteristics of different ecosystem types on false-color images, visual interpretation marks were established [46,47]. ArcGIS 10.6 was used for visual interpretation to obtain five LULC classes from 1992 to 2018, with a spatial resolution of 30 m (Figure 2). Next, the high-resolution data of Google Earth and field verification were combined for revision to ensure the accuracy of LULC data [46]. The overall classification accuracy meets the requirements of ecosystem research.

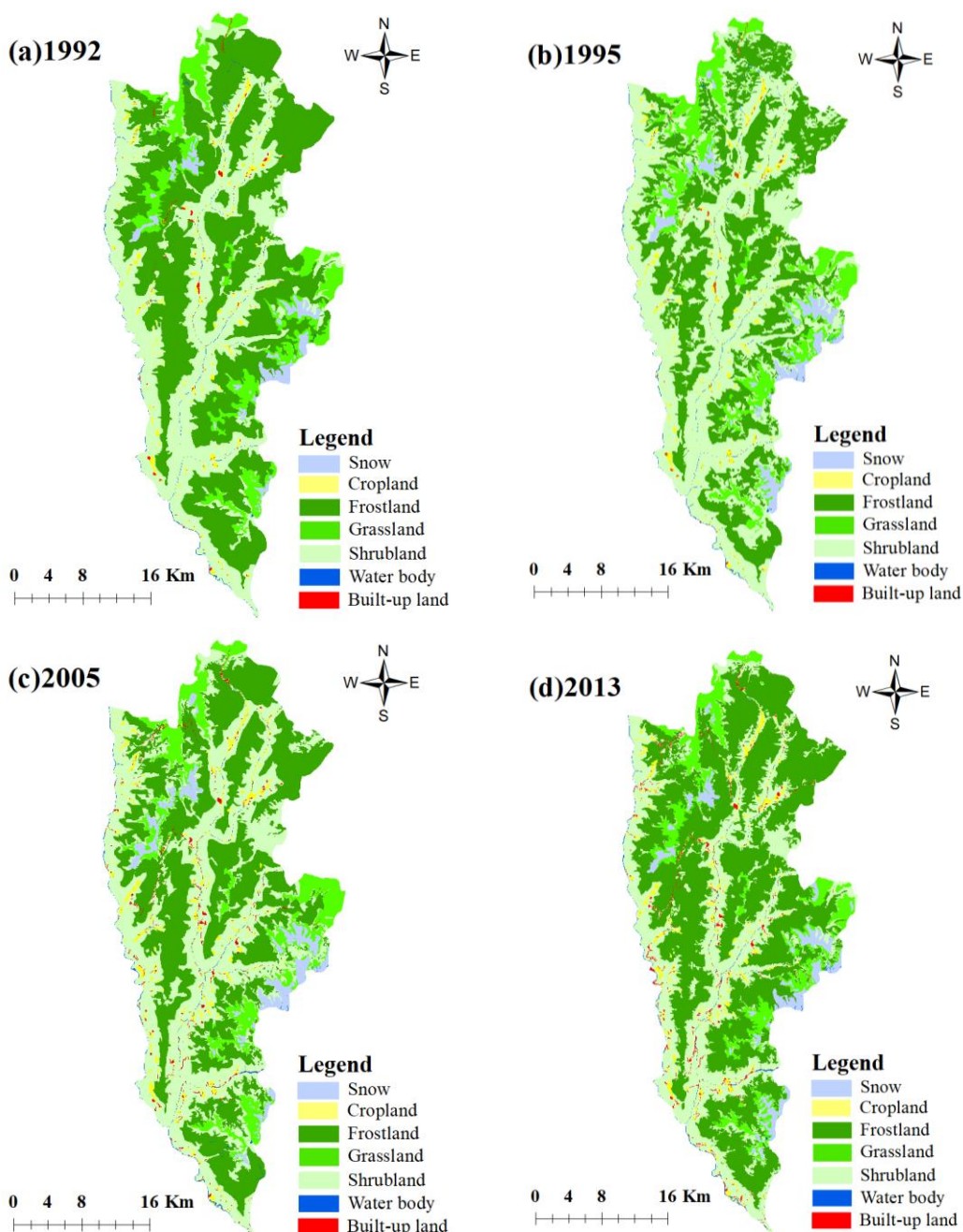

**Figure 2.** *Cont.*

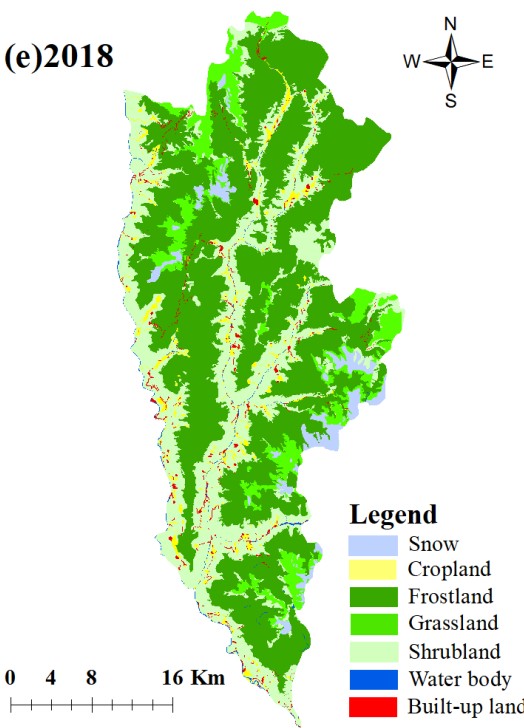

**Figure 2.** LULC maps of Derong County from 1992 to 2018. (**a**) 1992, (**b**) 1995, (**c**) 2005, (**d**) 2013, (**e**) 2018.

### 2.3. LULC Change Dynamics

The single land-use dynamics (M) refers to the annual variability rate of a certain type of land-use area within a concrete time range, which can better analyze land-use changes [34], calculated as follows:

$$M = \frac{V_1 - V_0}{V_0} \times \frac{1}{T_1 - T_0} \times 100\% \tag{1}$$

where $V_0$ and $V_1$ were the area (hm$^2$) of a given type of land-use in the starting and end years of the study period, and $T_0$ and $T_1$ were the starting and end years of the study time, respectively.

### 2.4. ESVs Assessment

Based on the ESV assessment system and historical literature, each land-use type included 11 ecosystem service function types [1–3]. The equivalent coefficients of ESV referred to Costanza and Xie [1–3,48], and according to historical literature and the "Ecosystem Service Value Equivalent Scale for China's Terrestrial Ecosystem" by Xie Gaodi [48–50], this study revised the ecosystem service value equivalent of Derong County. Meanwhile, given the negative and positive impacts of built-up land on the ecosystem, the built-up land estimation coefficients were developed according to the actual situation in Derong County, and the results are shown in Table 1.

**Table 1.** The ecosystem service value (ESV) equivalent coefficients of land-use/land cover change (LULC) in Derong County.

| Ecosystem Service | Forest Land | Shrub Land | Grass Land | Water Body | Crop Land | Built-Up Land | Snow |
|---|---|---|---|---|---|---|---|
| Food production | 0.22 | 0.19 | 0.22 | 0.80 | 1.08 | 0.00 | 0.00 |
| Raw material | 0.52 | 0.43 | 0.33 | 0.23 | 0.26 | 0.00 | 0.00 |
| Water supply | 0.27 | 0.22 | 0.18 | 8.29 | −1.17 | −0.02 | 2.16 |
| Gas regulation | 1.70 | 1.41 | 1.14 | 0.77 | 0.87 | −0.04 | 0.18 |
| Climate regulation | 5.07 | 4.23 | 3.02 | 2.29 | 0.45 | 0.00 | 0.54 |
| Purify environment | 1.49 | 1.28 | 1.00 | 5.55 | 0.13 | 0.10 | 0.16 |
| Hydrological regulation | 3.34 | 3.35 | 2.21 | 102.24 | 1.37 | −0.06 | 7.13 |
| Soil formation | 2.06 | 1.72 | 1.39 | 0.93 | 0.57 | −0.02 | 0.00 |
| Nutrient cycle | 0.16 | 0.13 | 0.11 | 0.07 | 0.15 | 0.00 | 0.00 |
| Biodiversity | 1.88 | 1.57 | 1.27 | 2.55 | 0.17 | 0.02 | 0.01 |
| Aesthetic landscape | 0.82 | 0.69 | 0.56 | 1.89 | 0.07 | 0.03 | 0.09 |
| Total | 17.53 | 15.22 | 11.43 | 125.61 | 3.95 | 0.01 | 10.27 |

Xie Gaodi et al. regarded the net profit of grain production per unit area of farmland ecosystem as a standard equivalent ecology factor and determined that the economic value of an ecosystem value equivalent in China in 2010 is 3406.50 yuan/hm² [1,44,49]. According to the average annual grain production per unit area in Derong County from 1992 to 2018 and the national grain per unit area yield (respectively 3083.26 kg/hm² and 4973 kg/hm²) [44], the farmland ecosystem service equivalent value coefficient in the study area was revised to 0.62, and finally the *ESV* of the unit equivalent factor in the research area obtained as 2112.03 yuan /hm²a. The *ESV* for each land-use type per hectare in different ecosystem services was calculated using Equation (2). The *ESV* calculation formula is as follows:

$$VC_i = \sum_{j=0}^{n} EC_j \times E_a \tag{2}$$

$$ESV = \sum_{i=0}^{n} A_i \times VC_i \tag{3}$$

where *ESV* is the value of ecosystem services, *i* is a land-use type, *j* is ecosystem service type, $A_i$ is the area of the class *i* land-use type (hm²), $VC_i$ is the *ESV* per unit area of class *i* land-use type (yuan/hm²a), $EC_j$ is the value equivalent of item *j* ecosystem services of a definite type of land-use, $E_a$ is the economic value of a unit ecosystem service value equivalent factor as 2112.03 (yuan/hm²a).

### 2.5. Elasticity of ESV Response to LULC

To eliminate the uncertainty caused by land-use type in *ESV* assessment, this study uses the conception of elasticity coefficient in economics to calculate the coefficient sensitivity (*CS*) of *ESV* to ensuring the reliability of research results. The *ESV* changes in response to 50% adjustments of the *ESV* coefficients for each LULC type were assessed [51]. The calculation formula of the *CS* is as follows:

$$CS = \left| \frac{(ESV_j - ESV_i)/ESV_i}{(VC_{jk} - VC_{ik})/VC_{ik}} \right| \tag{4}$$

where *ESV* is the estimated total value of ecosystem services, *VC* is the value coefficient, and "*i*," "*j*" and "*k*" represent the initial, adjusted values, and LUCC categories, respectively. If *CS* > 1, the estimated ESV is elastic concerning that coefficient. If *CS* ≤ 1, the estimated *ESV* is inelastic. Thus, when *CS* < 1, even if the accuracy of *VC* values used as proxy biomes is low, the results of the estimation are credible.

## 3. Results

### 3.1. Changes of LULC

The spatial distribution of LULC in Derong County from 1992 to 2018 was shown in Figure 2 and the dynamic change of the land-use area was shown in Table 2 and Figure 3. From 1992 to 2018, the change in LULC was significant. The overall characteristics of LULC: The primary LULC types in five periods were forestland and shrubland, which together account for more than 83% of the total area, followed by grassland and snow. In 1992, 2005, 2013, 2018, LULC was dominated by forestland, which accounted for 49.36%, 40.78%, 51.93%, and 54.98% of the total area, respectively. However, in 1995, due to deforestation, LULC was dominated by shrubland (47.21%), the forestland accounted for only 36.06%. In 1998, China implemented forestry protection projects and banned deforestation, which resulted in the increase of forest land in the region and the shift of people's lifestyle towards agriculture, which led to the growth of cropland.

**Table 2.** LULC area changes in Derong County from 1992 to 2018.

| LULC Year | | Forestland | Shrubland | Grassland | Water Body | Built-Up Land | Cropland | Snow |
|---|---|---|---|---|---|---|---|---|
| Area (hm²) | 1992 | 143,938.93 | 104,681 | 28,830.93 | 1481.87 | 911.17 | 4523.72 | 7232.39 |
| | 1995 | 105,156.57 | 137,652.87 | 31,022.1 | 1542 | 943.31 | 4671.79 | 10,611.35 |
| | 2005 | 118,904.84 | 123,697.47 | 27,532.9 | 1694.92 | 2642.73 | 6143.04 | 10,984.1 |
| | 2013 | 151,436.74 | 92,681.83 | 25,465.18 | 1560.1 | 3302.35 | 6345.81 | 10,807.99 |
| | 2018 | 160,309.32 | 83,277.34 | 27,209.29 | 1633.66 | 3590.85 | 6584.98 | 8994.56 |
| Changes (%) | 1992–2005 | −17.39 ↓ | 18.17 ↑ | −4.50 ↓ | 14.38 ↑ | 190.04 ↑ | 35.80 ↑ | 51.87 ↑ |
| | 2005–2018 | 34.82 ↓ | −32.68 ↓ | −1.18 ↓ | −3.61 ↓ | 35.88 ↑ | 7.19 ↑ | −18.11 ↓ |
| | 1992–2018 | 11.37 ↑ | −20.45 ↓ | −5.62 ↓ | 10.24 ↑ | 294.09 ↑ | 45.57 ↑ | 24.36 ↑ |

Note: "↑" means increase, and "↓" means decrease.

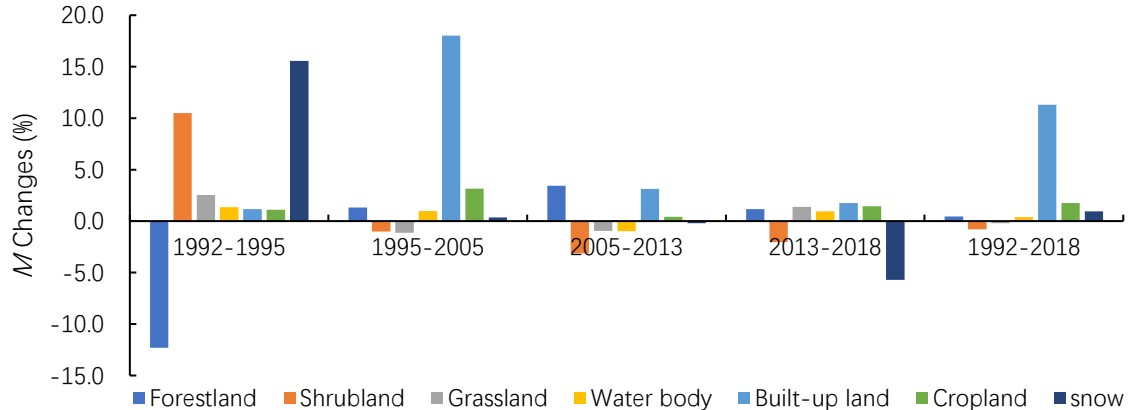

**Figure 3.** Single land-use dynamic (*M*) changes from 1992 to 2018.

The transformation trends of land-use types were as follows: Forestland first decreased and then increased, shrubland first increased and then decreased, grassland first increased and then decreased and then increased, water body first decreased and then increased, built-up land kept increasing, cropland kept increasing, and snow first increased and then decreased.

From 1992 to 2018, forestland, water body, built-up land, cropland, and snow increased, while shrubland and grassland decreased. With agricultural economic development and urban expansion, the built-up land and cropland increased significantly. The built-up land has changed the most, increasing by 2679.68 hm² and with an increase rate of 294.09% by 1992. The cropland also increased by 2061.26 hm² and with an increase rate of 45.57% by 1992. The shrubland decreased most by 21403.66 hm² and with a decrease rate of 20.45% by 1992.

The results of land-use dynamics from 1992 to 2018 showed that shrubland and grassland decreased, with dynamics of −0.79% and −0.22%, respectively. The other land-use types increased, and the order of increasing speed was: Built-up land > cropland > snow > forestland > water body, with dynamic degrees of 11.31%, 1.75%, 0.94%, 0.44%, and 0.39%, respectively.

### 3.2. Changes of ESV

LULC change causes the variation of ecosystem service value. As can be seen from Table 3, from 1992 to 2018, the ESV in Derong County first showed a decrease, then increase and decrease trend, decreased by 0.25% overall. The water body, snow, forestland, and cropland played a vital role in improving ESV. However, the ESV increase did not offset the decrease in ESV in the shrubland and grassland system. The forestland ecosystem services increased by $6.23 \times 10^8$ yuan, with an increase rate of 11.37% by 1992, the ESV of water body increased by $0.41 \times 10^8$ yuan, with an increase rate of 10.24%, the ESV of snow increased by $0.39 \times 10^8$ yuan, with an increase rate of 24.36%, and the ESV of cropland increased by $0.17 \times 10^8$ yuan, with an increase rate of 45.57%. At the same time, the ESV of shrubland decreased to $6.82 \times 10^8$ yuan, with a decrease rate of 20.45%. The ESV of grassland decreased by $0.38 \times 10^8$ yuan, with a decrease rate of 5.62%.

**Table 3.** Changes of ESV for different LUCC from 1992 to 2018.

| LUCC | ESV ($\times 10^8$ yuan) | | | | | Changes Rate (%) | | |
|---|---|---|---|---|---|---|---|---|
| | 1992 | 1995 | 2005 | 2013 | 2018 | 1992–2005 | 2005–2018 | 1992–2018 |
| Forestland | 53.29 | 38.93 | 44.02 | 56.07 | 59.35 | −17.39 ↓ | 34.82 ↑ | 11.37 ↑ |
| Shrubland | 33.65 | 44.25 | 39.76 | 29.79 | 26.77 | 18.17 ↑ | −32.68 ↓ | −20.45 ↓ |
| Grassland | 6.96 | 7.49 | 6.65 | 6.15 | 6.57 | −4.50 ↓ | −1.18 ↓ | −5.62 ↓ |
| Water body | 3.93 | 4.09 | 4.50 | 7.11 | 4.33 | 14.38 ↑ | −3.61 ↓ | 10.24 ↑ |
| Built-up land | 0.00 | 0.00 | 0.00 | 0.00 | 0.00 | 190.04 ↑ | 35.88 ↑ | 294.09 ↑ |
| Cropland | 0.38 | 0.39 | 0.51 | 0.53 | 0.55 | 35.80 ↑ | 7.19 ↑ | 45.57 ↑ |
| snow | 1.57 | 2.30 | 2.38 | 2.34 | 1.95 | 51.87 ↑ | −18.11 ↓ | 24.36 ↑ |
| Total | 99.78 | 97.45 | 97.82 | 102.00 | 99.53 | −1.96 ↓ | 1.74 ↑ | −0.25 ↓ |

From the perspective of the ESV composition of each LULC type, forestland, shrubland, and grassland were the three LULC types that contributed the most to the ESV composition, accounting for more than 88% of the total system value (Figure 4).

Soil and water conservation, the implementation of forest projects, and comprehensive ecological management have promoted the restoration of forestland and water body, and promoted the increase in the ESV of forestland and water body. Agricultural economic development and urban expansion have led to the reduction of shrubland and grassland areas and associated ESVs.

As can be seen from Table 4, from 1992 to 2018, climate regulation, hydrological regulation, soil conservation, and biodiversity contributed the most to the ESV, with a contribution of 71.34%, 71.51%, 71.53%, 69.56%, and 71.34%, most of ecosystem service functions decreased except food production, water supply, and hydrological regulation. From 1992 to 2005 and from 2005 to 2018, all individual ecosystem service functions showed the opposite trend (Figure 5).

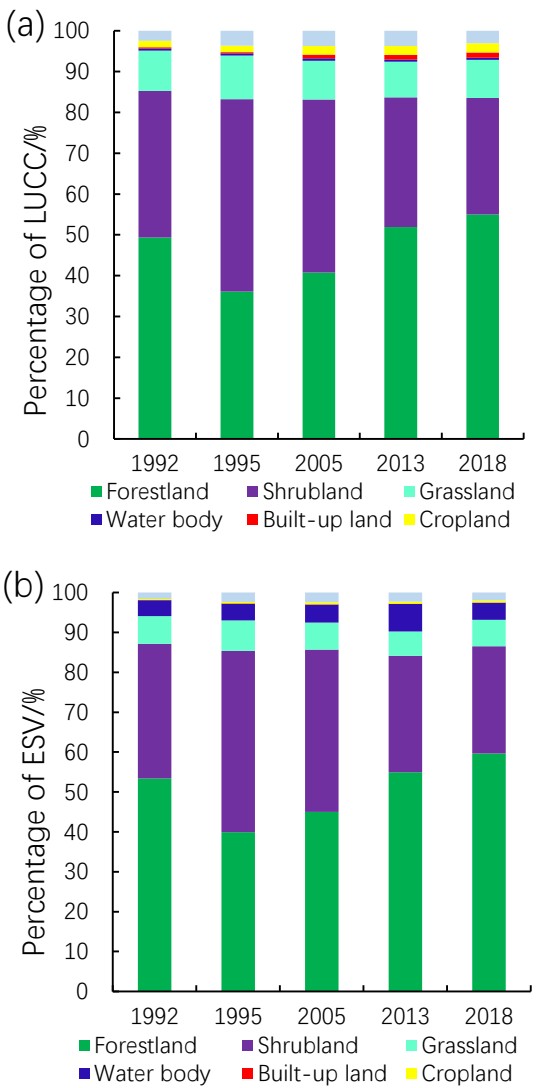

**Figure 4.** The percentage of LULC and ESV of different land-use types. (**a**) Percentage of LULC, (**b**) Percentage of ESV.

**Table 4.** Changes of ESV components in Derong County from 1992 to 2018.

| Ecosystem Service | ESV (×10⁸ yuan) | | | | | Changes Rate (%) | | |
|---|---|---|---|---|---|---|---|---|
| | 1992 | 1995 | 2005 | 2013 | 2018 | 1992–2005 | 2005–2018 | 1992–2018 |
| Food production | 1.35 | 1.32 | 1.35 | 1.36 | 1.38 | −0.41 ↓ | 2.80 ↑ | 2.38 ↑ |
| Raw material | 2.76 | 2.65 | 2.66 | 2.79 | 2.75 | −3.67 ↓ | 3.29 ↑ | −0.50 ↓ |
| Water supply | 1.89 | 2.00 | 2.00 | 2.00 | 1.94 | 5.72 ↑ | −3.28↓ | 2.25 ↑ |
| Gas regulation | 9.11 | 8.77 | 8.80 | 9.22 | 9.07 | −3.49 ↓ | 3.11 ↑ | −0.48 ↓ |
| Climate regulation | 26.80 | 25.78 | 25.81 | 26.38 | 26.59 | −3.72 ↓ | 3.02 ↑ | −0.80 ↓ |
| Purify environment | 8.18 | 7.92 | 7.93 | 9.70 | 8.12 | −3.12 ↓ | 2.43 ↑ | −0.77 ↓ |
| Hydrological regulation | 23.32 | 23.67 | 23.91 | 23.61 | 23.54 | 2.52 ↑ | −1.57 ↓ | 0.92 ↑ |
| Soil conservation | 10.99 | 10.57 | 10.58 | 11.08 | 10.91 | −3.76 ↓ | 3.09 ↑ | −0.78 ↓ |
| Nutrient cycle | 0.86 | 0.82 | 0.83 | 0.85 | 0.86 | −3.49 ↓ | 3.56 ↑ | −0.05 ↓ |
| Biodiversity | 10.06 | 9.67 | 9.68 | 10.64 | 9.97 | −3.77 ↓ | 3.03 ↑ | −0.86 ↓ |
| Aesthetic landscape | 4.44 | 4.28 | 4.29 | 4.37 | 4.41 | −3.44 ↓ | 2.78 ↑ | −0.75 ↓ |
| Total | 99.78 | 97.45 | 97.82 | 102.00 | 99.53 | −1.96 ↓ | 1.74 ↑ | −0.25 ↓ |

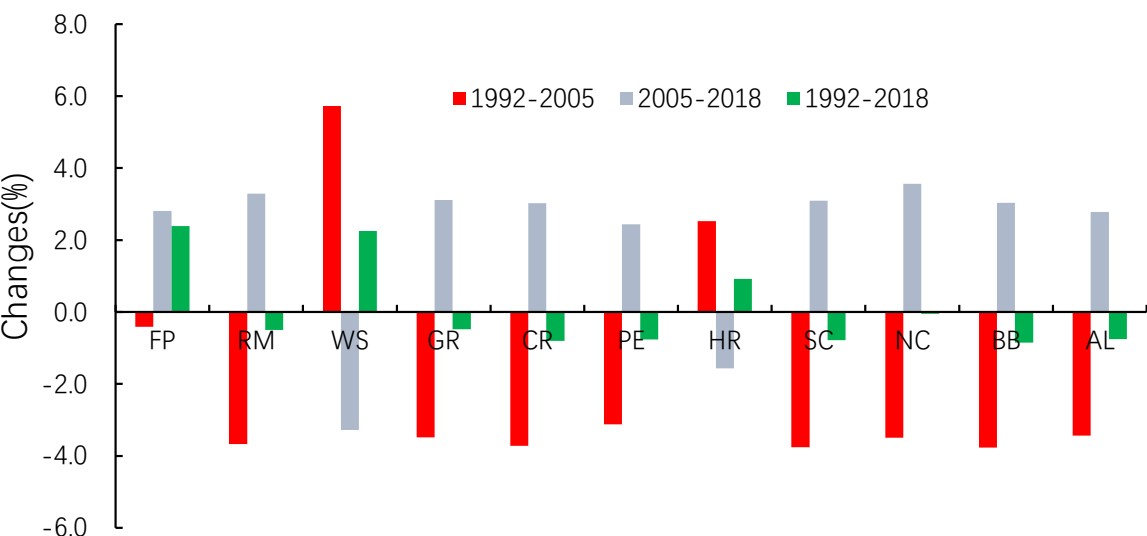

**Figure 5.** The change rate of ecosystem service function in Derong County from 1992 to 2018. FP, Food production, RM, Raw material, WS, Water supply, GR, Gas regulation, CR, Climate regulation, PE, Purify environment, HR, Hydrological regulation, SC, Soil conservation, NC, Nutrient cycle, BB, Biodiversity, AL, Aesthetic landscape.

Ranked by total value of service functions, they were climate regulation > hydrology regulation > soil conservation > biodiversity > gas regulation > purify environment > water supply > aesthetic landscape > raw material > food production > nutrient cycle (Figure 6).

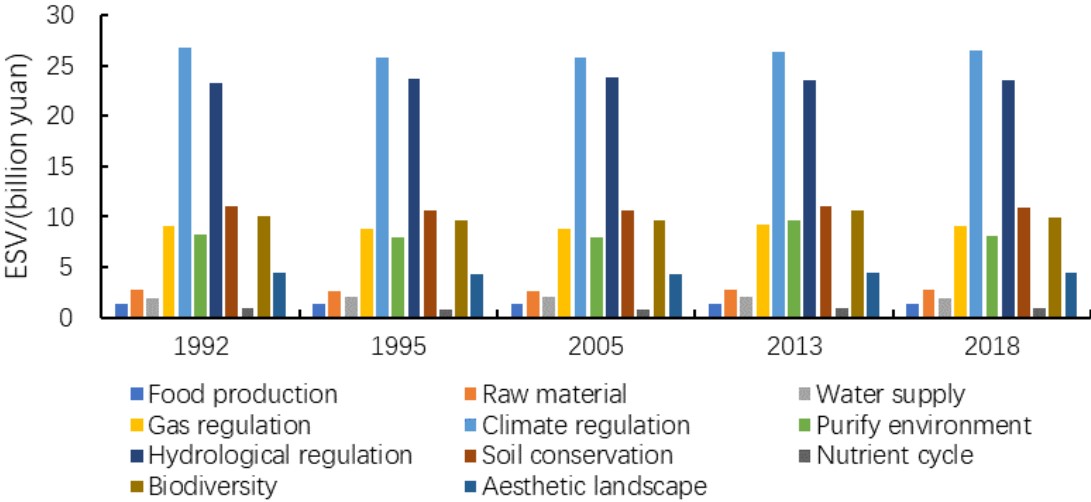

**Figure 6.** Ecosystem service functions and value composition of Derong County in 1992–2018.

The ESV spatial distribution in Derong County was unbalanced (Table 5). The high ESV area is largely distributed in Ciwu, Baisong, Bari and Bendu, their ESV exceeds 10 billion yuan. The low ESV area is mainly distributed in Rilong, Sizha, Xulong and Songmai, their ESV is less than 6 billion yuan (Figure 7). From Figure 7 and Table 6, 1992–2018, the value density of ESV was on the decline, and the most severe decrease in the value density of ESV was in Dianyagong. The highest density of ESV area is distributed in Waka, and the lowest density of ESV area is distributed in Bari.

**Table 5.** Changes of ESV in different townships regions from 1992 to 2018.

| LUCC | ESV (×10^8 yuan) | | | | | Changes Rate (%) | | |
|---|---|---|---|---|---|---|---|---|
| | 1992 | 1995 | 2005 | 2013 | 2018 | 1992–2005 | 2005–2018 | 1992–2018 |
| Xulong | 5.81 | 5.64 | 5.75 | 6.09 | 5.76 | −1.04 ↓ | 0.27 ↑ | −0.78 ↓ |
| Guxue | 9.51 | 9.20 | 9.33 | 9.66 | 9.51 | −1.91 ↓ | 1.94 ↑ | 0.00 ↓ |
| Bendu | 10.82 | 10.90 | 10.35 | 11.48 | 11.24 | −4.32 ↓ | 8.65 ↑ | 3.96 ↑ |
| Baisong | 11.98 | 11.67 | 11.81 | 12.15 | 11.96 | −1.38 ↓ | 1.22 ↑ | −0.17 ↓ |
| Songmai | 5.54 | 5.44 | 5.53 | 5.68 | 5.49 | −0.22 ↓ | −0.58 ↓ | −0.80 ↓ |
| Dianyagong | 7.42 | 7.30 | 7.46 | 7.60 | 7.10 | 0.55 ↑ | −4.86 ↓ | −4.34 ↓ |
| Ciwu | 13.69 | 13.31 | 13.21 | 13.62 | 13.56 | −3.48 ↓ | 2.62 ↑ | −0.95 ↓ |
| Gongbo | 8.18 | 7.75 | 7.99 | 8.23 | 8.1 | −2.27 ↓ | 1.44 ↑ | −0.86 ↓ |
| Sizha | 4.94 | 4.83 | 4.86 | 5.01 | 4.98 | −1.63 ↓ | 2.39 ↑ | 0.73 ↑ |
| Rilong | 3.53 | 3.46 | 3.53 | 3.70 | 3.52 | 0.04 ↑ | −0.23 ↓ | −0.19 ↓ |
| Waka | 7.13 | 7.05 | 7.25 | 7.50 | 7.19 | 1.78 ↑ | −0.81 ↓ | 0.96 ↓ |
| Bari | 11.25 | 10.90 | 10.75 | 11.29 | 11.10 | −4.40 ↓ | 3.24 ↑ | −1.30 ↓ |
| Total | 99.78 | 97.45 | 97.82 | 102.00 | 99.53 | −1.96 ↓ | 1.74 ↑ | −0.25 ↓ |

**Table 6.** Changes of ESV density in different townships regions from 1992 to 2018.

| LUCC | ESV Density (yuan/hm²) | | | | | Changes (yuan/hm²) | | |
|---|---|---|---|---|---|---|---|---|
| | 1992 | 1995 | 2005 | 2013 | 2018 | 1992–2005 | 2005–2018 | 1992–2018 |
| Xulong | 34,374.97 | 33,369.16 | 34,019.98 | 36,016.61 | 34,079.15 | −354.99 ↓ | 59.17 ↑ | −295.83 ↓ |
| Guxue | 33,464.17 | 32,373.33 | 32,830.78 | 33,980.97 | 33,464.17 | −633.39 ↓ | 633.39 ↑ | 0.00 ↓ |
| Bendu | 32,850.95 | 33,093.84 | 31,423.97 | 34,843.31 | 34,126.13 | −1426.98 ↓ | 2702.16 ↑ | 1275.18 ↑ |
| Baisong | 35,440.86 | 34,523.78 | 34,937.94 | 35,952.84 | 35,381.69 | −502.92 ↓ | 443.75 ↑ | −59.17 ↓ |
| Songmai | 34,718.83 | 34,092.14 | 34,656.16 | 35,626.22 | 34,405.49 | −62.67 ↓ | −250.68 ↓ | −313.35 ↓ |
| Dianyagong | 35,260.08 | 34,689.83 | 35,450.16 | 36,111.65 | 33,739.43 | 190.08 ↑ | −1710.73 ↓ | −1520.65 ↓ |
| Ciwu | 33,966.09 | 33,023.28 | 32,775.17 | 33,789.67 | 33,643.55 | −1190.92 ↓ | 868.38 ↑ | −322.54 ↓ |
| Gongbo | 33,334.78 | 31,582.46 | 32,560.50 | 33,531.24 | 33,049.52 | −774.28 ↓ | 489.02 ↑ | −285.26 ↓ |
| Sizha | 34,291.70 | 33,528.12 | 33,736.37 | 34,793.87 | 34,569.37 | −555.33 ↓ | 833.00 ↑ | 277.67 ↑ |
| Rilong | 35,664.08 | 34,956.86 | 35,664.08 | 37,340.72 | 35,563.05 | 0.00 ↑ | −101.03 ↓ | −101.03 ↓ |
| Waka | 36,567.48 | 36,157.19 | 37,182.93 | 38,450.35 | 36,875.21 | 615.44 ↑ | −307.72 ↓ | 307.72 ↑ |
| Bari | 33,191.59 | 32,158.96 | 31,716.41 | 33,295.42 | 32,749.04 | −1475.18 ↓ | 1032.63 ↑ | −442.55 ↓ |
| Total | 34,217.81 | 33,420.10 | 33,547.49 | 34,977.82 | 34,130.94 | −670.32 ↓ | 583.45 ↑ | −86.87 ↓ |

### 3.3. Ecosystem Sensitivity Analysis

After adjusting the value factors of ecosystem services for each land-use type by 50%, respectively (Table 7), the results showed that the sensitivity index of the five periods of 1992 to 2018 was less than 1, indicating that the value of ecosystem services used in this study was inelastic, and the results were credible.

**Table 7.** The sensitivity coefficients in Derong County during 1992–2018.

| CS | 1992 | 1995 | 2005 | 2013 | 2018 |
|---|---|---|---|---|---|
| Forestland | 0.1335 | 0.0999 | 0.1125 | 0.1374 | 0.1491 |
| Shrubland | 0.0843 | 0.1135 | 0.1016 | 0.0730 | 0.0672 |
| Grassland | 0.0174 | 0.0192 | 0.0170 | 0.0151 | 0.0165 |
| Water body | 0.0098 | 0.0105 | 0.0115 | 0.0174 | 0.0109 |
| Built-up land | 0.0000 | 0.0000 | 0.0000 | 0.0000 | 0.0000 |
| Cropland | 0.0009 | 0.0010 | 0.0013 | 0.0013 | 0.0014 |
| Snow | 0.0039 | 0.0059 | 0.0061 | 0.0057 | 0.0049 |

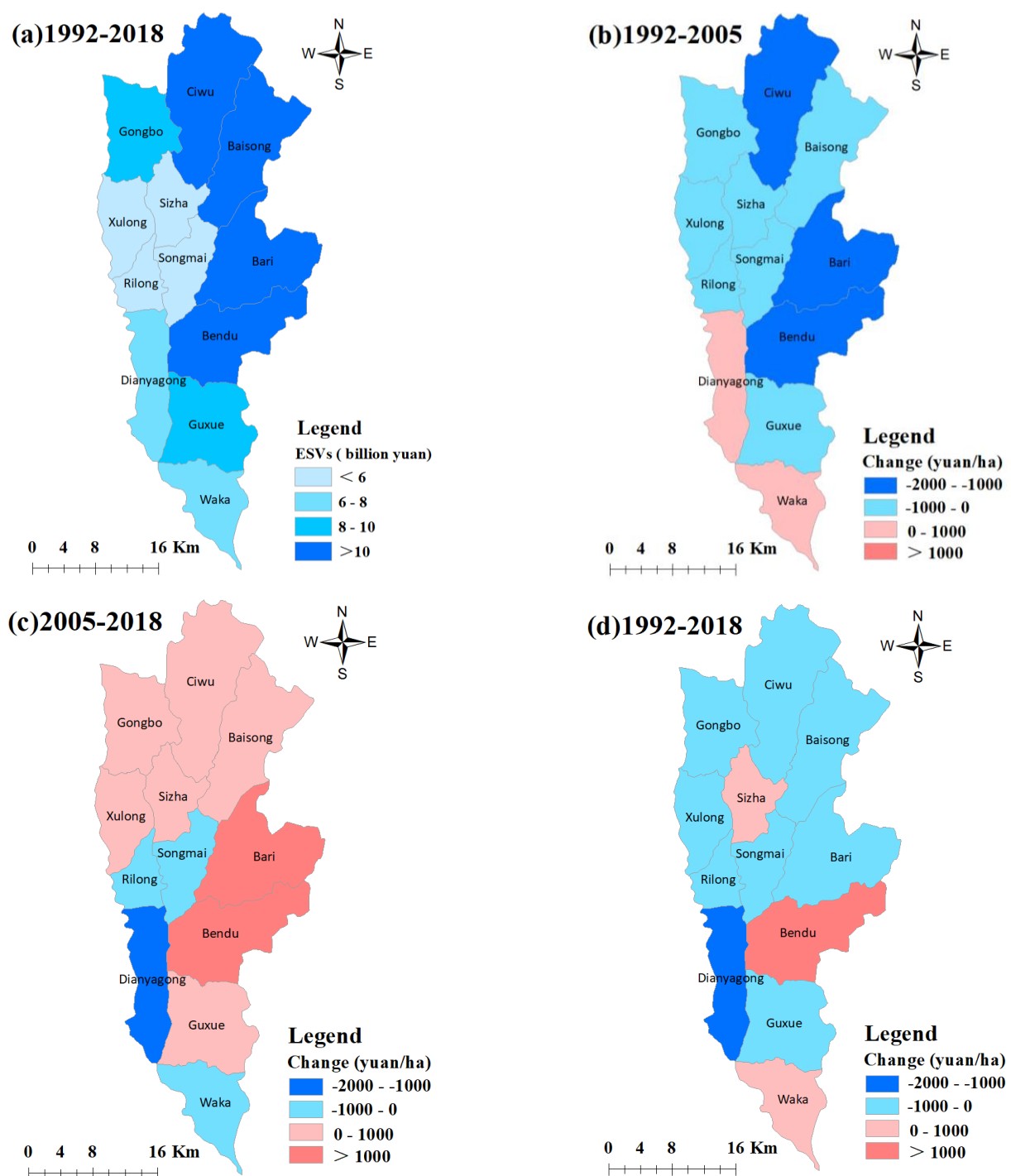

**Figure 7.** Spatial patterns of ecosystem service values (ESVs) (**a**) and changes in ESVs density at the townships region level during (**b**) 1992–1995, (**c**) 1995–2005, and (**d**) 1992–2018.

## 4. Discussion

### 4.1. Effects of LULC Changes on ESV

LULC changes directly affect ESV [52]. Optimizing land-use structure and protecting natural ecosystems will help increase the ESV. Ecological protection projects are critical to increase the ESV [23]. In this study, deforestation has been shown to have a gigantic impact on the ESV, during 1992–1995, the forestland decreased by 38782.36 hm$^2$ and ESV decreased by $14.36 \times 10^8$ yuan. During 1995–2018, the natural forest protection projects in the upper reaches of the Yangtze River played a key role in improving the

ESV, the forestland increased by 55,152.75 hm$^2$ and ESV increased by $20.42 \times 10^8$ yuan. From 1992–2018, although ecological protection projects had been implemented, urban expansion and agricultural economic development caused the ESV of the Derong County to decline by $0.25 \times 10^8$ yuan. Many studies have quantified and evaluated the impact of rapid urbanization on ecosystem services. Although the methods for assessing ecological and economic characteristics are different, the results consistently showed that in the current urban development, ecosystem service functions tend to decline. Therefore, how to manage complex and diverse ecosystem services in the process of rapid urbanization is a big challenge for sustainable development [37,53,54]. This research showed that to provide local communities with better ecological benefits and continue to improve the ecological environment of the upper reaches of the Yangtze River, it is necessary to reasonably implement ecological projects, optimize land-use structure, and control the forestland-shrubland ratio [41]. The coordinated development of economic development and ecological protection is essential for achieving sustainable development [55,56]. Therefore, future land planning should focus on the close relationship between humans and ecosystems, such as attaching importance to the combined application of landscape measurement analysis and ESV assessment, advocating the development of multi-center cities, long-term monitoring changes in landscape patterns, considering the supply of ecosystem services and the spatial interaction of ecological land [38,57], and paying more attention to environmental protection and natural protection, this will help guide future human activities.

This study conducts a temporal and spatial quantitative study on the ESV of Derong County, which has certain theoretical and practical significance for restoring and protecting the ecosystem of Derong County and accelerating the comprehensive management of the ecological environment in Derong County. The research results help people more intuitively understand the importance of ecosystem services in Derong County, and at the same time, provides a scientific reference for the effective protection and management of ecosystems in the area and the sustainable use of land resources. The intuitive monetary value highlights the importance of Derong County's land resources to the regional economy and ecology and also provides a scientific theoretical basis for the protection and management of land ecosystems in countries around the world.

### 4.2. Limitations and Future Work

The ESV assessment methods used in this study can be used for rapid spatial analysis, assessment, and quantification of long-term ecological benefit restoration. This research is of great significance to the optimization of regional land-use patterns, maintenance of regional ecological security, and sustainable development. It is of enormous meaning for the ecological protection and construction of the upper reaches of the Yangtze River, especially for the ecological construction of dry-hot valley regions. Meanwhile, this study provides a case for assessing the impacts of the LULU on ESV, which has a certain reference value for the formulation of land-use policies. Next, we will conduct research on the impact of other factors (such as altitude, climate change) on the ESV. The shortcomings of the current research are as follows: (1) This study ignores the influence of altitude on land-use and ESV. (2) LULC classification was affected by the spatial resolution of remote sensing data.

### 5. Conclusions

With the support of Landsat TM/OLI remote sensing data and GIS technology, this study conducted a quantitative analysis of the temporal and spatial changes in ESV caused by LULC in Derong County from 1992 to 2018. The major study results showed as follows: (1) From 1992 to 2018, the ESV with an overall decrease of $0.25 \times 10^8$ yuan, and the most severe decline in the value density of ESV was in Dianyagong. (2) Ecological projects have played a positive role in improving ESV in Derong county. (3) Of the 11 individual ecosystem service values, the value contribution rate of climate regulation was the largest. (4) The sensitivity indexes were all less than 1, and the results were reliable. To continue to

improve the ecological restoration in the upper reaches of the Yangtze River and achieve sustainable development in the dry-hot valley area, it is necessary to strengthen soil and water conservation, natural forest protection projects, and other ecological restoration measures, and to provide better ecological well-being for local communities.

**Author Contributions:** Data curation, Y.W.; Funding acquisition, P.P.; Investigation, Y.W.; Methodology, Y.W. and X.Z.; Project administration, P.P.; Resources, P.P.; Supervision, P.P.; Validation, X.Z.; Writing—original draft, Y.W. All authors have read and agreed to the published version of the manuscript.

**Funding:** This research was financially supported by the Biodiversity Survey and Assessment Project of the Ministry of Ecology and Environment, China (No. 2019HJ2096001006).

**Institutional Review Board Statement:** Not applicable.

**Informed Consent Statement:** Not applicable.

**Data Availability Statement:** Data available in a publicly accessible repository that does not issue DOIs.

**Acknowledgments:** We acknowledge the geospatial data cloud for the freely downloadable remote sensing images. We would like to thank the reviewers for their constructive and detailed comments.

**Conflicts of Interest:** The authors declare no conflict of interest.

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
