# Peer review of "Spatio-Temporal Changes of Land-Use/Land Cover Change and the Effects on Ecosystem Service Values in Derong County, China, from 1992–2018"

_sustainability, doi:10.3390/su13020827_

Round 1

Reviewer 1 Report

The work is timely and interesting – the research methodologies are reasonable, and the findings are justifiable. However, there are still a few aspects that should be improved. As the article is coherent and well-organised, I focus here only on a few major points, which are hopefully easy for the authors to take into account in the revision.

  • The implications for this research seem to be too limited to the local geographic areas. As noted by the authors “these results will provide a reference for other counties in the Hengduan Mountains region”, are there any wider implications to the international community. Since Sustainability is an international journal, the authors should think about why the wider audience without local knowledge can be interested in this work.

  • The local adaptations to the ESV coefficients of LULC in Derong County should be highlighted in Table 1. The current description (lines 106-118) is too simplified. More details should be given to any changes to the coefficients.

  • The price level in calculating ESV is not clear? Is this in the real price? If so, in which year?

  • The policy implications should be strengthened with more details (Lines 237-240). For instance, how to optimise the urban spatial structure to balance the social and environmental needs simultaneously (suggest reading, for instance, https://doi.org/10.1016/j.apgeog.2011.12.001 and https://doi.org/10.1177/2399808319864972)?

  • Despite many interesting findings, the theoretical contribution of this paper is not clearly identified. I recommend the authors adding a paragraph highlighting the theoretical contribution in Section 4 or 5.

Reviewer 2 Report

First of all, this manuscript needs a thorough English editing to improve readability. There are numerous grammatic errors (the reviewer noted some in the annotated pdf file attached).

More importantly, the reviewer is concerned with several foundational issues of this manuscript and the study. To begin with, the authors justify this research by pointing to the need of Derong County. In my opinion, it would not serve as a reasonable justification for carrying out research. The author should ground the work from more general theoretical perspectives, not on the need of a particular county.

Moreover, the methodology is heavily dependent on remote sensing data. However, in the Introduction, the authors never discuss remote sensing data (e.g., the opportunities it brings for ESV research).

In addition, the classification of remote sensing imagery into LULC type was purely visually-based. How reliable/accurate is it? Why not using supervised classification methods? The visually-based process is a very subjective process and prone to human errors. At least the authors did not provide enough evidence/justification to convince the reviewer that the method is reliable.

Reviewer 3 Report

See comments in attached review document. The conclusions should be expanded. Additionally, the LULC change methods could be better implemented directly in GoogleEarth Engine using Java scripting.

Round 2

Reviewer 1 Report

All my concerns have been addressed - I believe the current version can be published.

Reviewer 2 Report

Thanks to the authors for adequate addressing the reviewer's comments. The reviewer recommends publication.